# Discovery and Validation of a Novel Step Catalyzed by *OsF3H* in the Flavonoid Biosynthesis Pathway

**DOI:** 10.3390/biology10010032

**Published:** 2021-01-06

**Authors:** Rahmatullah Jan, Sajjad Asaf, Sanjita Paudel, Sangkyu Lee, Kyung-Min Kim

**Affiliations:** 1Division of Plant Biosciences, School of Applied Biosciences, College of Agriculture & Life Science, Kyungpook National University, 80 Dahak-ro, Buk-gu, Daegu 41566, Korea; rehmatbot@yahoo.com; 2Natural and Medical Science Research Center, University of Nizwa 616, Nizwa 611, Oman; sajadasif2000@gmail.com; 3College of Pharmacy, Research Institute of Pharmaceutical Sciences, Kyungpook National University, 80 Dahak-ro, Buk-gu, Daegu 41566, Korea; sanjitapdl99@gmail.com (S.P.); pharm239@gmail.com (S.L.); 4Department of Botany, Garden Campus, Abdul Wali Khan University, Mardan 23200, Pakistan; lubnabilal68@gmail.com

**Keywords:** kaempferol, naringenin, nuclear magnetic resonance, yeast episomal plasmid, hydroxylation

## Abstract

**Simple Summary:**

Flavonoids are important plant secondary metabolites mostly produced in the shikimate pathway. Kaempferol and quercetin are important anti-oxidant flavonoids, which enhance plant tolerance to environmental stresses. The biosynthesis of both the flavonoids largely depends on the expression of genes of the shikimate pathway. Therefore, we selected the *OsF3H* gene from rice and assessed its functional expression using the yeast expression system. We found that *OsF3H* regulates a very important step of the flavonoid biosynthesis pathway and enhances the accumulation of kaempferol and quercetin. The present research confirmed that overexpression of the *OsF3H* gene in rice could significantly increase the biosynthesis of flavonoids, which are essential for the plant defense system.

**Abstract:**

Kaempferol and quercetin are the essential plant secondary metabolites that confer huge biological functions in the plant defense system. In this study, biosynthetic pathways for kaempferol and quercetin were constructed in *Saccharomyces cerevisiae* using naringenin as a substrate. *OsF3H* was cloned into pRS42K yeast episomal plasmid (YEp) vector and the activity of the target gene was analyzed in engineered and empty strains. We confirmed a novel step of kaempferol and quercetin biosynthesis directly from naringenin, catalyzed by the rice flavanone 3-hydroxylase (*F3H*). The results were confirmed through thin layer chromatography (TLC) followed by western blotting, nuclear magnetic resonance (NMR), and liquid chromatography-mass spectrometry LCMS-MS. TLC showed positive results when comparing both compounds extracted from the engineered strain with the standard reference. Western blotting confirmed the lack of *OsF3H* activity in empty strains and confirmed high *OsF3H* expression in engineered strains. NMR spectroscopy confirmed only quercetin, while LCMS-MS results revealed that *F3H* is responsible for the conversion of naringenin to both kaempferol and quercetin.

## 1. Introduction

Flavonoids and isoflavonoids are essential plant aromatic secondary metabolites. They cover a very large family of phenolic compounds that mediate diverse biological functions and exert significant ecological impacts. The flavonoids class encompasses approximately 1000 known compounds [1], including anthocyanin, proanthocyanidin, and phlobaphene pigments, as well as flavonol, flavone, and isoflavone with their respective biological functions in host species [2,3,4]. As related to plant biological activities, flavonoids mostly play a role in the plant defense system, antimicrobial activity, UV light protection, and even through tissue-specific compartmentalization [5,6]. Researchers reported that some flavonoids are involved in auxin transport inhibition, have allelopathic activity, and regulate reactive oxygen species in plants [7]. Among flavonoids, kaempferol and quercetin are the predominant naturally occurring phenolic flavonol compounds with a common flavone nucleus. Both these compounds are of great importance because of their anti-cancer, cardio-protective, and anti-inflammatory activities [8]. Furthermore, they inhibit the growth of cancer cells, induce apoptosis of cancer cells, and preserve normal cell viability [9]. Kaempferol and quercetin are present in the glycoside form in nature and have numerous biological functions like antioxidant, anti-diabetes, anti-inflammatory, antimicrobial, and anti-fungal activities [10,11].

In plants, the kaempferol and quercetin biosynthetic pathways are well developed, and both kaempferol and quercetin are synthesized by complexes of various enzymes, via the phenylpropanoid pathway, from aromatic amino acids like phenylalanine and tyrosine [12]. At a very early step in the flavonoid biosynthesis pathway, phenylalanine is converted to naringenin through a series of reactions catalyzed by various enzymes. Naringenin is the main intermediate compound that acts as a precursor for the synthesis of various flavonoids, depending on the enzymes it interacts with [13]. Particularly, *F3H* catalyzes hydroxylation of flavanones at the 3 position of the C-ring and converts them into dihydroflavanoles [14]. Synthesis of flavonols (kaempferol and quercetin) from dihydrokaempferol and dihydroquercetin is catalyzed by flavonol synthase (*FLS*) [15]. Previous reports showed that kaempferol and quercetin are synthesized by the activation of different enzymes in different organisms. For example, the functional expression of naringenin 3-dioxygenase (*N3COX*) from *Petroselinum crispum*, in yeast, generates dihydrokaempferol, which is further converted into dihydroquercetin, kaempferol, and quercetin under the action of the respective enzymes [16], as shown in (Figure 1A). Similarly, cytochrome P450 flavonoid monooxygenase (*FMO*), which was fused in-frame to the cytochrome P450 reductase (*CPR*) from *Catharanthus roseus*, was expressed in yeast and produced quercetin from kaempferol [17] (Figure 1B). According to this mechanism, kaempferol is directly converted to quercetin under the action of of *FMO/CPR*. However, in arabidopsis, dihydrokaempferol is first converted into dihydroquercetin under the action of the *F3′H* enzyme (Figure 1C), which is further converted to quercetin by the *FLS* enzyme [18]. In lotus, dihydrokaempferol is converted into kaempferol under the action of dihydroflavonol reductase (*DFR*), whereas the enzyme responsible for the formation of quercetin is unknown [19] (Figure 1D). The proposed pathway of flavonols in *Rubus* shows that naringenin is hydroxylated by *F3H* with the formation of dihydrokaempferol, which is further converted into kaempferol under the action of *FLS*. Further conversion of kaempferol into dihydroquercetin, a precursor of quercetin, is catalyzed by the F3′H enzyme [20] (Figure 1E). Naringenin conversion into various flavonoids is achieved through the addition of a hydroxyl group to different positions of the compound, depending on the enzymes that catalyze the conversion [14]. Previously, Leonard et al. [21] and Miyahisa et al. [22] reported kaempferol synthesis in *Escherichia coli* using L-Tyrosin as a primary substrate and cloned the entire pathway, whereas Leonard, Yan and Koffas [21] used p-coumaric acid as the precursor of kaempferol and quercetin in *E.coli* and cloned only the downstream pathway. Flavanone 3′-hydroxylase (*F3′H*) also has a prominent role in flavonol biosynthesis and has 50% sequence similarity with *F3H*. It has been reported that quercetin is produced by the hydroxylation of kaempferol at the B3 position under the action of the F3′H enzyme [20]. It is confirmed that *F3′H* is not only responsible for the conversion of dihydrokaempferol into dihydroquercetin, but can also convert kaempferol into quercetin. The aim of our study is to validate the function of rice flavanone 3-hydroxylase enzyme with respect to the synthesis of dihydroflavanoles and flavonol.

## 2. Materials and Methods

### 2.1. Cloning and Construction of Plasmids

The construct for transformation to yeast was prepared in three steps via restriction-based cloning: (1) insert preparation, (2) construction of vector, and (3) ligation process. To prepare the insert, the gene (Os04g0662600, http://www.gramene.org/) was amplified with the forward primer “5′ggatccATGGCGCCGGTGGCCACGACGTT3′” with the BamH1 site and the reverse primer “5′ctcgagTCACTGCTCTGACGAAGCAACAGAGCAG3′” with the Xho1 site, and purified by gel electrophoresis using the Qiagen QIAquick Gel Extraction Kit (Cat # 28706). The purified insert was treated with BamH1 and Xho1 restriction enzymes (New England BioLabs) via incubation at 37 °C for 4 h. To prevent methylation, 40 µL of the insert was treated with 2 µL Dpn1 enzyme in the presence of 1µl Cutsmart buffer (10×) (New England BioLabs) for 2 h at 37 °C. At the same time, pRSk42 vector was also treated with BhamH1 and Xho1 enzymes for 4 h at 37 °C. After cutting with restriction enzymes, the vector was treated with calf intestine phosphatase (CIP) enzyme (New England BioLabs) to reduce the chances of phosphorylation of vector ends. In the last step, the insert was ligated to the vector in the ratio of 5:1, respectively, in the presence of Quick Ligase enzyme and 2× Quick Ligase Reaction buffer (New England BioLabs). Before transfer to the yeast, the construct was transformed to *E. coli* JM109 cells, via the heat shock method, as pRS42k plasmid amplifies in both *E.coli* as well as yeast cells but expresses only in yeast cells. The transformation was confirmed by the isolation of the plasmid DNA from *E.coli*, digestion with restriction enzymes, and the observation of two expected bands of the vector and target gene.

### 2.2. Strain and Media

*E. coli* DH5α and *S. cerevisiae* D452-2 were the strains used in this experiment, having pRS42k plasmid with PGK1 promoter and CYC1 terminator in both *E.coli* and yeast. YPD media (1% yeast extract, 2% peptone, and 2% glucose) were used as the basal media for the routine growth of yeast described by [23]. After autoclaving the solid and liquid media, 150 mg/L of Geneticin (G418) and Spectinomycin (Invitrogen) were added as selection markers after cooling.

### 2.3. Transformation to Yeast

*S. cerevisiae* transformation was carried out via the lithium acetate/single-stranded carrier DNA/polyethylene glycol (LiAc/SS carrier DNA/PEG) method [24]. The yeast strain was grown overnight in 10mL YPD medium at 30 °C in a shaker at 200 rpm; further, a 250 mL YPD culture flask was incubated. After 12–14 h incubation, the titer of yeast culture was determined by adding 10 µL cells into 1 mL water in a spectrophotometer cuvette and observing the optical density (OD) at 600 nm. Subsequently, 2.5 × 10^8^ cells were added to 50 mL of pre-warmed YPD into a pre-warmed flask with the titer of 5 × 10^6^. The flask was incubated at 30 °C and 200 rpm for about 4 h, after which the cells were harvested and washed with 30 mL water and again rewashed in 1 mL water. Finally, the cells were resuspended in 1 mL of water by vortexing. Simultaneously, single-stranded carrier DNA (Salmons perm DNA, Sigma Chemical Co. Ltd., St Louis, MO, USA, cat. no. D-1626) was incubated for 5 min in a boiling water bath for denaturation and chilled on ice immediately. PEG 50% *w*/*v* (50g of PEG 3350 added to 30 mL of sterile water) and LiAC 1.0M (10.2 g LiAC autoclaved with 100mL water for 20 min) were prepared. Following this, 360 µL transformation mix (35 µL plasmid, 36 µL LiAC, 240 µL PEG, and 50 µL of SS carrier DNA) was added to 100 µL of transformation tube (yeast cells) and vortexed vigorously. The cells were incubated at 42 °C for 40 min in a water bath and then harvested by centrifugation for 30 s at 13,000 rpm. The supernatant was discarded and the cells were resuspended in 1 mL distilled water by pipetting and 40 µL of cells were plated on each selection media. The plates were incubated for 3 days at 30 °C and the transformants were counted.

### 2.4. Colony PCR

Following the method described by Ling, et al. [25], fresh yeast colonies were used directly for PCR without plasmid purification to confirm gene transformation. Template DNA was prepared by adding a single colony picked using a sterile toothpick to 30 µL of 0.03M NaOH; the mixture was vortexed vigorously and boiled for 3 min. This was followed by centrifugation at 5000 rpm for 1 min. The supernatant was discarded and 2 µL of the pellet was used as template DNA. This method is highly accurate compared with the old method of adding yeast directly to PCR tubes. All the PCR conditions were the same as those described previously except the initial denaturation time, which was set at 5 min. PCR products were evaluated on 1% gel and target DNA was visualized using a gel manager.

### 2.5. Protein Isolation and Western Blot Analysis

The protein was isolated from yeast strains according to the method described by [26], with a minor modification. Yeast strain (10 mL) was collected in a 50-mL falcon tube and centrifuged at 5000 rpm for 5 min at 4 °C. The supernatant was discarded and the pellet was resuspended in 5 mL TEK buffer solution (50 mM Tris pH 7.5, 2 mM EDTA, and 100 mM KCl) and centrifuged again at 5000 rpm for 5 min. The pellet was resuspended in 5 mL TES buffer solution (50 mM Tris pH 7.5, 2 mM EDTA, 0.8 M sorbitol, 20 mM mercaptoethanol, and 2 mM phenylmethylsulfonyl fluoride) and disrupted by bead beating. Then, 140 mM PEG3350 and 0.2 g/mL NaCl were added to the supernatant and the mixture was incubated on ice for 15 min. After incubation, the samples were centrifuged for 10 min at 10,000 rpm and the pellet was resuspended in 100 µL of TEG solution (50 mM Tris pH 7.5, 2 mM EDTA, and 40% glycerol). Protein concentrations were determined by Bradford method [27]. The isolated protein (20 µg) was loaded for separation on 12% polyacrylamide gel as described by Laemmli [28]. After separation on polyacrylamide gel, the protein was electro-transferred to nitrocellulose membrane (using transfer apparatus from Bio-Red) and it was suspended in a blocking buffer (50 mM Tris-Cl pH 7.4, 150 mM NaCl, 0.1% Tween 20, and 5% skim milk) for 90 min at room temperature, similar to the study by [29]. After washing with tris buffered saline-tween (TBST) (50 mM Tris-Cl pH 7.4, 150 mM NaCl, and 0.1% Tween 20) for 40 min, the membrane was incubated in a corresponding primary antibody in a 1/1200 dilution and polyclonal anti-mouse IgG antibody from goat (Invitrogen) as a secondary antibody at room temperature. Immunodetection was performed using enhanced chemiluminescence (ECL) Western Blotting Detection Reagents (Amersham, Buckinghamshire, UK) and an Image Quant™ LAS 4000 system (Fujifilm, Tokyo, Japan).

### 2.6. Extraction and Thin Layer Chromatography (TLC) of Kaempferol and Quercetin

Kaempferol and quercetin were extracted using 100 mL liquid culture grown for 7 days. The crude extracts were isolated using different extraction chemicals; their ratios and extraction time are given in Appendix A. Crude extracts were then separated into different fractions using a 100-cm-long silica gel cylinder. The fractions were dried in a rotary evaporator and dissolved in methanol and ethanol for further analysis on TLC. TLC was performed according to the standard method described by [30], with minor modifications. The fresh solvent was used for each run as they are volatile and also kept at room temperature because increasing the temperature increases evaporation of the solvent. Approximately 5 µL of 1 mg/mL extracts and the same amount of standard were loaded on the TLC plate (20 × 20 cm thickly coated with 0.4–0.5 nm silica gel) and dried with a blow dryer; the extracts were then allowed to run in their respective mobile phases in the TLC chamber for 25 min. The developed plates were fully dried for 20 min at room temperature and then directly visualized in the TLC viewer under 366 nm UV light. The same process was repeated in triplicate. The detected bands were matched with the reference compounds of kaempferol and quercetin. The matching bands were crumb and collected individually and eluted with methanol for further assistance.

### 2.7. Kaempferol and Quercetin Identification via LCMS-MS and NMR

The samples separated through fractionation for TLC analysis were further identified with LCMS-MS. A quantity of 3 µL of each sample was mixed with 79 µL of 50% CAN (0.1% formic acid), and 2 µL was injected to perform LCMS-MS analysis. All LC/MS analyses were performed using a linear trap quadrupole (LTQ) Orbitrap XL (Thermo Electron Co., Madison, WI, USA) coupled to an Accelar ultra-high-pressure liquid chromatography system (Thermo, Waltham, MA, USA). The LCMS-MS were carried out in three technical repeats. Chromatographic separation of metabolites was conducted using an ACQUITY UPLC^®^ BEH C18 column (2.1 × 150 mm, 1.7 µm, Waters, Milford, MA, USA), operated at 40 °C, and using mobile phases A (water) and B (acetonitrile with 0.1% formic acid) at the flow rate of 0.4 mL/min.^1^H and ^13^C NMR spectra were recorded on a Bruker Avance II 400 (Bruker, Billerica, MA, USA) in methanol-d4 solutions. Working frequencies were 400.1 and 101.0 MHz for ^1^H and for ^13^C, respectively.

### 2.8. Statistical Analysis

Experiments were performed in triplicate and the values obtained are presented as the means ± standard deviation (SD). Data obtained ware statistically evaluated by duncan’s multiple range test (DMRT) using GraphPad Prism (version 6.01, San Diego, CA, United States).

## 3. Results and Discussion

### 3.1. Cloning Of F3H and Designing of Kaempferol and Quercetin Pathway

In recent years, many synthetic tools have been developed with regard to yeast engineering to produce value added secondary metabolites. The proposed pathway for biosynthesis of kaempferol and quercetin is depicted in (Figure 1). Previous reports showed that *F3H* is a major member of the flavonoid biosynthesis enzymes found in all organisms, which enhance kaempferol and quercetin biosynthesis [31]. Additionally, researchers have overexpressed whole genes related to the flavonoid pathway in yeast [32]. The precursor of the flavonoid pathway, p-coumaric acid, is synthesized either from tyrosine in tyrosine ammoni-lyase (TAL), or from phenylalanine in the phenylalanine ammonialyase (PAL) pathway [17]. Furthermore, it is demonstrated that p-coumaric acid is activated by a 4-coumaroyl-CoA ligase (*4CL*) from *P. crispum*, while chalcone synthase (*CHS*) from *P. hybrid,* and chalcone isomerase (*CHI*) from *M. sativa* were used to convert the resulting p-coumaroyl-CoA into naringenin [17]. Many downstream metabolite synthesis pathways merge from naringenin depending on the activation enzymes. However, depending on the requirement, researchers have cloned the entire pathway, or a target step of the flavonoid pathway, in the microbial factories [16].

In this study, we used restriction-based cloning of the complete open reading frame (ORF) region of the *OsF3H* gene ligated into the pRS42k expression vector between the PGK1 promoter and CYC1 terminator (Appendix A). We first transformed the construct to *E. coli* for rapid and efficient multiplication, high copy numbers, and confirmation of ligation. Transformation and ligation were confirmed by plasmid isolation and digestion with corresponding restriction enzymes (Appendix A). The same construct isolated from *E. coli* was further transformed to *S. cerevisiae* for functional expression. Approximately 70% of the transformations were successful as 7 colonies were transformed out of 10 and the transformation was confirmed through colony PCR (Appendix A) and the selection marker. Multiple copy numbers are essential to expresses heterologous genes in yeast; however, significant multiplication of heterologous genes puts pressure on cells and results, which may cause uncertainty of the construct. To maintain the stability and optimum copy number of constructs, yeast episomal plasmid (YEp) vector was selected, due to its autonomous replication because of the presence of a 2-micron region that acts as the origin of replication. The 2-micron origin enhances the copy number, resulting in significant transformation, as most of the YEp plasmids range from 5 to 30 copies per single cell [33,34]. The engineered strain was induced by subjecting naringenin as a substrate for the *OsF3H* gene, and the resulting titer was further analyzed using biotechnological tools for confirmation of *OsF3H* gene expression, and synthesis of kaempferol and quercetin.

### 3.2. TLC Analysis

TLC is one of the most important procedures to confirm the presence of related compounds in the extract by comparing the extract with the reference. To confirm the formation of kaempferol and quercetin, TLC was performed using the standard method [30]. After fractional distillation and rotary evaporation, a minute quantity (1 mg/mL) of extracts of different profiles was dissolved in their relative solvents along with the same concentration of standard kaempferol and quercetin. TLC profiling was used to indicate the most prominent extract by comparing the extracted kaempferol and quercetin with the standards. Different chromatographic systems (Table 1) of different solvents, as a mobile phase, were used in random concentrations and a favorable solvent that clearly separated kaempferol and quercetin was selected. Different solvent profiles’ analysis results confirmed that toluene:ethylacetate:formic acid (7:3:0.5) separated kaempferol and quercetin significantly, as shown in Figure 2A. These results also revealed that there was no difference between the Rf of standard and that of the extracted sample. However, the Rf of kaempferol was found to be larger than that of quercetin in all the mobile phases. Similarly, quercetin appeared before kaempferol, showing that quercetin is more polar than kaempferol [35]. The bands that appeared on the TLC plate authenticated that flavanone 3-hydroxylase enzyme catalyzed the conversion of naringenin to kaempferol and quercetin.

### 3.3. OsF3H Expression in Yeast

To further confirm *OsF3H* activity and protein expression level, the transformed yeast as well as empty yeast (having an empty pRS42K vector) was grown for 24 h at 30 °C. After 24 h, naringenin was added to the cultured media of both the transformed and empty strain and proteins were isolated at three time points: 2 h, 12 h and 24 h. The functional expression of *OsF3H* was determined by the level of recombinant protein expression among the three time points via western blotting. Further, immunoblotting was performed with equal protein volumes by creating a standard curve to estimate the expression level of the recombinant protein. Our results confirmed that the empty vector does not produce the target protein that is similar to transformed strain, which shows a lack of *OsF3H* activity. Thus, it was proved that *S. cerevisiae* has no *F3H* activity and only transformation with *OsF3H* is responsible for the recombinant protein production. On comparing the expression level at each time point, it was assumed that the expression levels slightly increased after each interval because of the continuous catalytic activity of the enzyme (Figure 2B). This phenomenon predicts that continuous availability of the substrate increases the enzymatic activity of the *OsF3H* gene because of constant conversion of naringenin into its product. Selection of a suitable and applicable recombinant protein expression system plays a key role in the protein expression technique. It is known that the promoter is the most characterized genetic segment in many yeast expression systems [36]. Thus, to achieve a high-level protein expression, we selected a well characterized promoter PGK1 and CYC1 terminator to control the target protein expression of cDNA in the host cell. It has been reported that heterologous protein production and high degree transferability commonly relied on promoters of *S. cerevisiae* like PGK1 and terminators such as CYC1t [36]. Selection of a weak promoter can be discouraged due to low levels of transcription of foreign genes, and consequently, the production of low amounts of recombinant protein. Similarly, the selecting of a strong promoter is also not recommended all the time due to high levels of transcription of foreign genes, which can cause a severe stress and can lead to cell to death in case of unfolded protein response (UPR). In such cases, demands on protein folding and protein size are fundamental for choosing a proper promoter. In this regard, we reviewed previous literature for the most frequently used promoter and terminator, and selected the PGK1 and CYC1t promoter and terminator, respectively, and no damage was found—thus, it is possible to make a strong prediction that the target genes were expressed in a controlled circumstance.

### 3.4. Identification of Kaempferol and Quercetin via Nuclear Magnetic Resonance (NMR)

NMR spectroscopy was used to identify a compound by the demonstration of type, number, and position of atoms in a molecule. It is a comprehensive study involving the renovation of the chemical structure of a compound architecture by generating detailed information about carbon and hydrogen atoms in the structure. Currently, NMR application is one of the most useful tools for structural analysis of flavonoids. In the current study, we used NMR spectroscopy to further identify our target compound through structural evaluation. After TLC identification, the engineered strain fractions were further analyzed for identification by ^1^H and ^13^C NMR (Figure 3A,B, respectively). Deuteriochloroform is a common solvent used for direct NMR analyses of many flavonoids, isoflavones, flavones, and flavonols. We used Dichloromethane D from Cambridge Isotope Laboratories (CIL), USA, as most naturally occurring flavonoids (all flavonoid glycosides) are poorly soluble in deuteriochloroform. However, they are highly soluble in methanol, ethanol, dichloromethane and their derivatives. Samples subjected to NMR analysis revealed only a single compound; the structure was illustrated as quercetin. ^1^H-NMR (600 MHz) δH: 6.41 (1H, d, J = 2.4 Hz), 6.20 (1H, d, J = 2.4 Hz), 7.75 (1H, d, J = 3.0 Hz), 6.91 (1H, d, J = 9.6 Hz), 7.66–7.64 (1H, dd, J =9.6, 3.0 Hz). 13C-NMR (125 MHz) δC: 148.3 (C-2), 137.5 (C-3), 177.6 (C-4), 162.7 (C-5), 99.5 (C-6),165.9 (C-7), 94.7 (C-8), 158.5 (C-9), 104.8 (C-10), 124.4 (C-1′), 116.3 (C-2′), 146.5 (C-3′), 148.3 (C-4′), 116.5 (C-5′), 121.9 (C-6′). It was recognized as quercetin through comparison with the spectroscopically analyzed data reported in the literature [37]. The identification of quercetin and the lack of kaempferol implies that kaempferol is produced in a very low quantity because NMR detects a very decent amount and the highest purity level of compounds. Furthermore, sometimes a single enzyme can catalyze one step of the reaction more efficiently than the other step. It is also possible that the expression of the *OsF3H* gene in yeast significantly converts naringenin into dihydrokaempferol and dihydroquercetin, and at the same time, also converts the intermediates into quercetin only. The illustrated structure shows the hydroxyl groups attached on the 3′, 4′, 5, and 7 (Figure 3C) positions.

### 3.5. In Vivo Activity of OsF3H in Yeast and Quantification of Kaempferol and Quercetin via LCMS-MS

The LC-MS profiles of the *OsF3H* assay are shown in Figure 4. The enzymatic activity of different enzymes expands naringenin metabolism to different pathways that are responsible for the synthesis of diverse classes of flavonoids. Substrate conversion to kaempferol and quercetin is caused by the addition of single and double hydroxyl groups to naringenin, which is catalyzed by *F3′H* [14]. Our finding shows that using naringenin as a substrate, the recombinant yeast expressing *OsF3H* gene successfully metabolized naringenin directly into kaempferol and quercetin, which is a novel step discovered and elucidated for the first time (Figure 1F). As we only expressed *OsF3H* in the strain and produced kaempferol and quercetin, therefore, our study claimed that *OsF3H* might be bi-functional, which means that it first converts naringenin into dihydrokaempferol and then dihydrokaempferol is converted into kaempferol. Furthermore, the control strain with an empty vector showed the lack of kaempferol and quercetin accumulation, indicating that *OsF3H* is responsible for naringenin conversion to both compounds. Furthermore, our results did not claim that there is no FLS and F3′H activity in the DH42K strain, but it could possibly be negligibly less active because neither kaempferol nor quercetin synthesis was detected in empty strain. Results showed that metabolism of naringenin increased after each time point, which decreased the naringenin and increased kaempferol and quercetin significantly (*p* > 0.05) (Figure 4A). LC-MS results also indicated that *OsF3H* metabolized naringenin to kaempferol more significantly than to quercetin. The transformed strain synthesized 8.10 ± 3.81 mg/L, 12.13 ± 5.04 mg/L, and 19.13 ± 1.29 mg/L of kaempferol and 5.42 ± 1.25 mg/L, 7.56 ± 0.86 mg/L, and 8.67 ± 0.39 mg/L of quercetin after 2, 12 and 24 h, respectively. The concentration of naringenin in the titer also decreased gradually along with the increases in both metabolites, showing that *OsF3H* strongly catalyzed the reaction (Figure 4A).

Our evaluation and analysis showed that the *OsF3H* gene is responsible for catalyzing naringenin metabolism directly to kaempferol and quercetin without the involvement of *FLS*; chemical structures of all the three metabolites, naringenin, kaempferol and quercetin, are shown in Appendix A. Previously, the conversion of naringenin to dihydrokaempferol, via *F3H*, was reported, which is further converted to kaempferol via flavonol synthase and, at last, kaempferol is converted to quercetin via *F3′H* [38,39] in higher plants. Similarly, Marín, Gutiérrez-del-Río, Entrialgo-Cadierno, Villar and Lombó [16] recently reported that naringenin is a substrate metabolized to dihydrokaempferol in the presence of naringenin 3-dioxygenase (*N_3_DOX*), which is further converted to kaempferol and also to dihydroquercetin via *FLS* and *F3H*, respectively. Additionally, it has been reported that the silencing of *FLS* in plants reduces quercetin production, suggesting that without *FLS,* quercetin can also be produced [40], which is probably due to *F3H* activity. The production of quercetin without *FLS* activity validates that not only *FLS* is responsible for the hydroxylation of dihydroquercetin, which converts to quercetin, but some other enzymes are also responsible for the addition of the OH groups. The flavonoid biosynthesis pathway affirms that naringenin is converted to kaempferol and quercetin by hydroxylation, depending on enzyme specificity, by using O_2_ and nicotinamide adenine dinucleotide phosphate (NADPH) [21]. Our finding suggests that the *OsF3H* gene is responsible for the addition of the OH group to naringenin, dihydrokaempferol, and dihydroquercetin for synthesizing both kaempferol and quercetin.

## 4. Conclusions

In the current study, we used the *S. cerevisiae* D452-2 strain and pRS42k plasmid with the frequently used PGK1 promoter and CYC1t terminator. Here, we cloned the rice *OsF3H* gene that is in the yeast expression vector and is functionally expressed in yeast. It is confirmed that *OsF3H* is a bi-functional enzyme that converts naringenin into kaempferol and quercetin without expression of the *FLS* and *F3′H* enzymes. With regard to future prospects, this study may help to overexpress the *OsF3H* gene and assist multifunctional secondary metabolites’ flavonol biosynthesis in rice plants.

## Figures and Tables

**Figure 1 biology-10-00032-f001:**
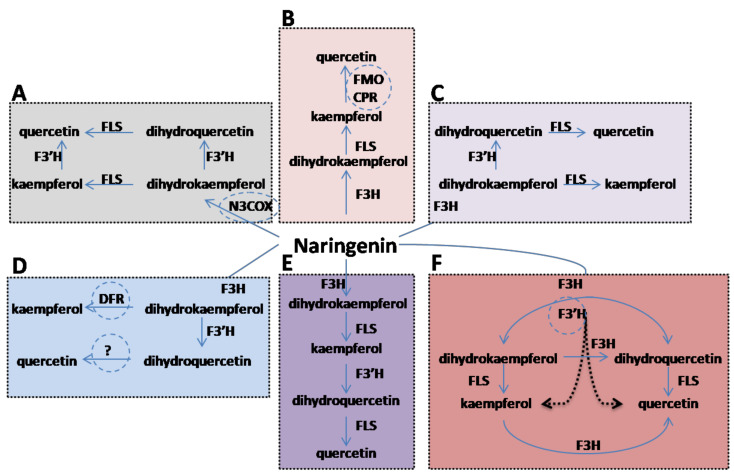
Schematic representation of the flavonoid biosynthetic pathway proposed for significant production of kaempferol and quercetin in different organisms. Naringenin is a substrate used by all the organisms using different enzymes for conversion to their respective products. (**A**) represents the proposed pathway in yeast using the *N3COX* gene converting naringenin into dihydrokaempferol. (**B**) shows the flavonoid pathway developed in microbes with the expression of the *FMO* and *CPR* gene catalyzing kaempferol into quercetin. (**C**) the flavonoid pathway in Arabidopsis shows that *F3′H* catalyzes dihydrokaempferol into dihydroquercetin while, (**D**) in Lotus, *DFR* converts dihydrokaempferol into kaempferol, instead of the FLS enzyme. (**E**) shows *F3′H* converting kaempferol into dihydroquercetin in blackberry and (**F**) is our proposed pathway for flavonoid synthesis in yeast using the rice *F3H* gene.

**Figure 2 biology-10-00032-f002:**
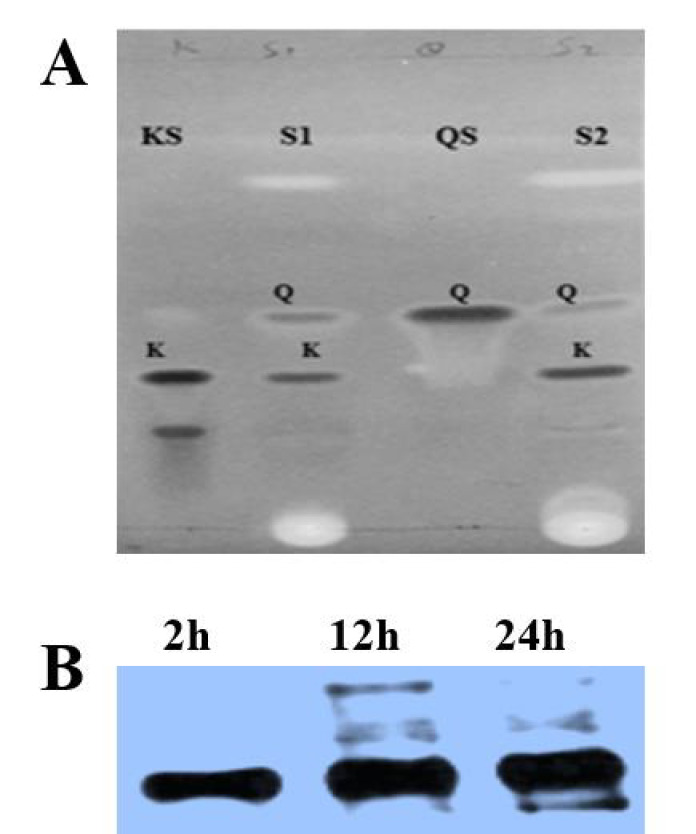
TLC analysis and functional expression of *OsF3H* gene in yeast. (**A**) TLC detection of kaempferol and quercetin under 366 nm UV light. Two samples were analyzed after fractionation through silica column, indicated as S1 and S2, mean sample one and two, respectively. At the same time, standards were also run for comparison, represented by KS as kaempferol standard and QS as quercetin standard, while K represents kaempferol and Q represents quercetin bands. (**B**) immunoblot analysis of yeast recombinant protein of the *OsF3H* gene. The same amounts of transformed strain (each time point) and empty strain protein were prepared and subjected to immunoblot analysis. SDS-PAGE analysis did not detect the target protein in the empty strain and it was not further analyzed for immunoblotting analysis. The time points of 2, 12 and 24 hrs, respectively, represent protein extracted from yeast strain feeding on substrate after each time point.

**Figure 3 biology-10-00032-f003:**
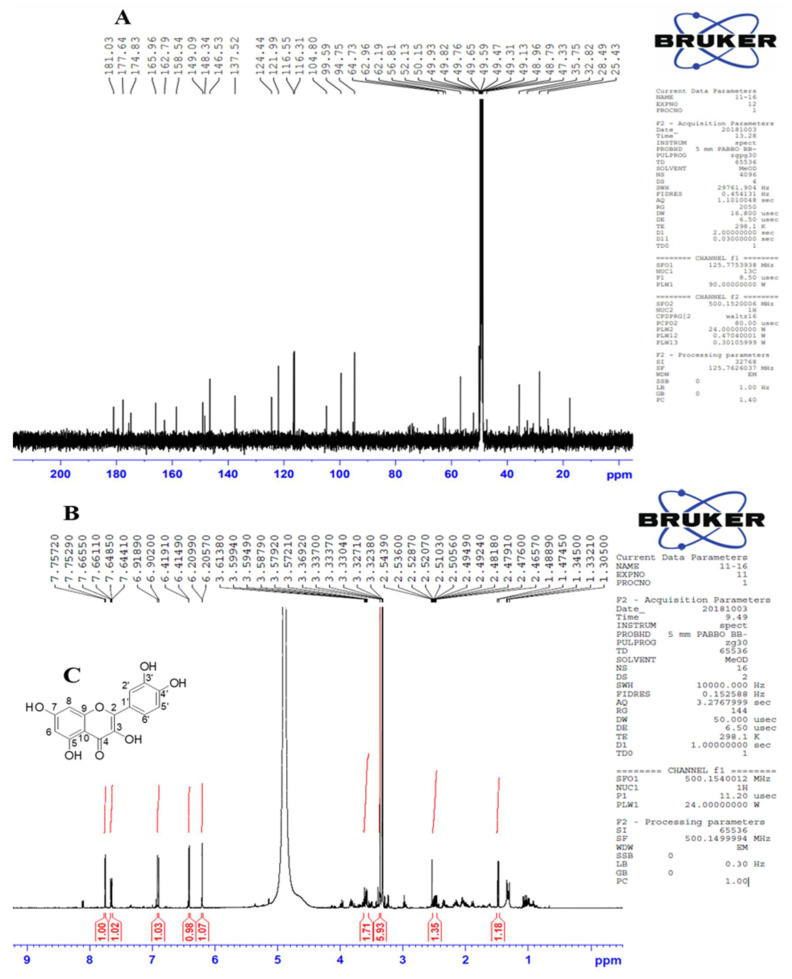
13C and 1H NMR spectra (**A**,**B**), respectively) of quercetin. The proposed structure is shown in (**C**).

**Figure 4 biology-10-00032-f004:**
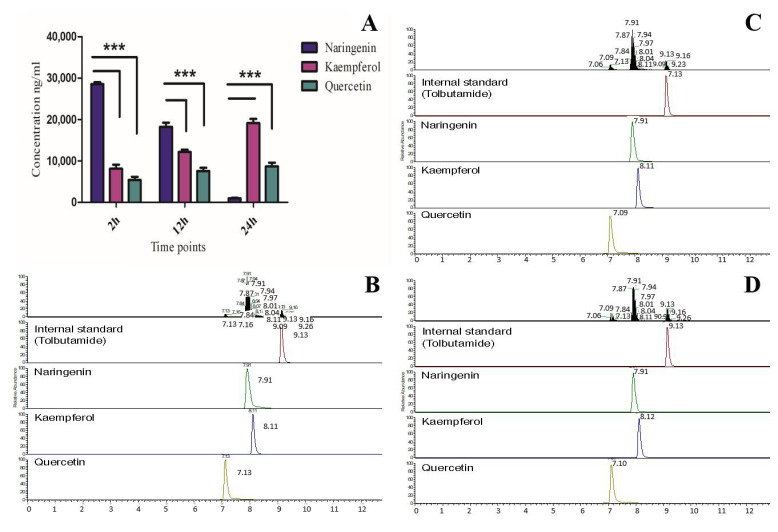
Analysis of naringenin conversion to kaempferol and quercetin using an engineered yeast system expressing the *OsF3H* gene by using LCMS-MS. (**A**) LCMS-MS profiling of naringenin, kaempferol and quercetin. Bars represent mean ± standard deviation, asterisks indicate significant difference (*p* < 0.05 two-way ANOVA, Bonferroni post-test) between naringenin, kaempferol and quercetin after each time point. (**B**–**D**) extracted ion chromatograms of internal standard (naringenin, kaempferol and quercetin) in sample extracted after 2, 12 and 24 h induction, respectively, in *OxF3H* yeast.

**Table 1 biology-10-00032-t001:** Optimized mobile phases, ratio of solvents used for TLC, and retention factor (Rf) values of kaempferol and quercetin.

Chromatographic System	Solvent	Ratio	Rf Value of Kaempferol	Rf Value of Quercetin	References
1	Ethyl acetate: glacial acetic acid: formic acid: water	100:11:11:25	0.49	0.35	[41]
2	Benzene: acetic Acid: water	125:72:3	0.35	0.24	[42]
3	N-butanol: acetic acid: water	4:01:05	0.45	0.31	[42]
4	Toluene: ethyl acetate: formic acid	5:04:01	0.36	0.24	[43]
5	N-hexane: ethylacetate: acetic acid	31:14:5	0.28	0.18	[43]
6	Toluene: ethyl acetate: formic acid	7: 3: 0.5	0.44	0.32	[44]

## Data Availability

All data generated or analyzed during this study are included in this published article.

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
