# Peer review of "Discovery and Validation of a Novel Step Catalyzed by OsF3H in the Flavonoid Biosynthesis Pathway"

_biology, 2021, doi:10.3390/biology10010032_

Round 1

Reviewer 1 Report

The authors cloned rice OsF3H gene in yeast expression vector and evaluated its functional expression and activity. They confirmed that OsF3H is a bi-functional enzyme convert naringenin into kaempferol and quercetin in a novel step. The authors suggested that OsF3H gene was responsible for the addition of OH group to naringenin, dihydrokaempferol, and dihydroquercetin for synthesizing both kaempferol and quercetin. The study is technically designed, the results are reliable. I retain that this manuscript could be accepted for publication, after the following minor revision.

Minor revision:

1.   Page 9-10, LC/MS results clearly indicated that OsF3H metabolized naringenin to kaempferol more significantly than to quercetin. Figure 4. A showed that OsF3H catalyzed the reaction. If possible, the authors should discuss kinetics of enzyme catalysis in this experiment.

Author Response

Response to Reviewer 1 Comments

  1. Page 9-10, LC/MS results clearly indicated that OsF3H metabolized naringenin to kaempferol more significantly than to quercetin. Figure 4. A showed that OsF3H catalyzed the reaction. If possible, the authors should discuss kinetics of enzyme catalysis in this experiment.

Response 1: Thank you for your important comment. We used the product of same reaction for both NMR and LC/MS analysis. We repeated NMR 3 times and identified quercetin but unfortunately, we did not clearly identify kaempferol that might be due to the poor quality of kaempferol. However, we further relied on LC/MS for identification and quantification of both the compound and it was clearly detected that both the compounds are resent in the same samples we used for NMR. This inference evaluated that F3H significantly catalyzed naringenin into kaempferol and quercetin.

Reviewer 2 Report

In this paper, the authors report a novel biosynthetic pathway in yeast of bioactive flavonoids, which are represented by kaempferol and quercetin. It is quite unfortunate for the reviewer that the description of chemical aspect, which is actually an essential part of the paper, is just poorly described, which would discourage readers to pick up the importance of the paper. Therefore, the reviewer would NOT recommend publication in the journal. For future improvements, obvious difficulties are listed as the followings.
(1) Although the manuscript focuses on conversion of chemical structures, it contains only trivial names of chemicals but no structures. This makes the details of manuscript hard to understand.
(2) The reviewer suggests that the authors ask a proofreading mainly for English description. There are some immature errors such as follows. The verb "synthesize" should be transitive. The Rf values can be large, not high. There are also a couple of misleading expressions that are related to active/passive voice.
(3) The reviewer also recommends that the authors ask an organic chemist to check description since there are a number of immature expressions such as follows. Flavonoids are typically solis or crystals at room temperature so that fractional distillation, which appears at "TLC analysis" section, would not be chosen when it comes to their purification. Silica gel column chromatography should not abbreviated unless they are clearly declared. TLC experiments require solvent conditions. The NMR spectrum in Figure 3 is obviously 1H NMR of 500 MHz, which is not properly labeled. These kinds of errors are found around the manuscript and really confusing.

Author Response

Response to Reviewer 2 Comments

In this paper, the authors report a novel biosynthetic pathway in yeast of bioactive flavonoids, which are represented by kaempferol and quercetin. It is quite unfortunate for the reviewer that the description of chemical aspect, which is actually an essential part of the paper, is just poorly described, which would discourage readers to pick up the importance of the paper. Therefore, the reviewer would NOT recommend publication in the journal. For future improvements, obvious difficulties are listed as the followings.

(1) Although the manuscript focuses on conversion of chemical structures, it contains only trivial names of chemicals but no structures. This makes the details of manuscript hard to understand.

Response 1: Thank you for your important comment. Actually, in our study we did not evaluate chemical structure and chemical aspect of related compounds because, the main focus of our study is related to the functional evaluation of OsF3H gene which is validated by different means. Kaempferol and quercetin are structurally and functionally well-known compounds.  

(2) The reviewer suggests that the authors ask a proofreading mainly for English description. There are some immature errors such as follows. The verb "synthesize" should be transitive. The Rf values can be large, not high. There are also a couple of misleading expressions that are related to active/passive voice.

Response 2: Thank you for your suggestion. The current manuscript is already edited by the English native; however, we removed the mistakes point out by the reviewer. Synthesizing were changed to “synthesis” line # 379 and High changed to “large” Line# 255. However, we find the word synthesize as appropriate word. Further we provided the English editing certificate.

(3) The reviewer also recommends that the authors ask an organic chemist to check description since there are a number of immature expressions such as follows. Flavonoids are typically solid or crystals at room temperature so that fractional distillation, which appears at "TLC analysis" section, would not be chosen when it comes to their purification. Silica gel column chromatography should not abbreviated unless they are clearly declared. TLC experiments require solvent conditions. The NMR spectrum in Figure 3 is obviously 1H NMR of 500 MHz, which is not properly labeled. These kinds of errors are found around the manuscript and really confusing.

Response 3: According to the reviewer suggestion we discussed with the expert organic chemist and we are agreed with the reviewer comments that flavonoids are solid in nature in pure form but it can be dissolved in various solvents. In this experiment we used methanol and ethanol for extraction and further re-dissolved both of the compounds in methanol after rotary evaporation. Both the compounds are separated on silica column in dissolved form because in methanol they cannot remain in crystal form. The same method is followed by various researchers for instance reviewer may check the paper of Sumit Arora [1].

We included some well-known conditions of solvent for TLC experiment in Line number 183-185. However, the choice of a suitable solvent depends on nature of substance, and adsorbent used on the plate. A development solvent should be such that, does not react chemically with the substances in the mixture under examination. As TLC is very commonly used for flavonoids so we followed very common conditions. We also added solvent in the NMR section in line # 310, 313. Further we will appreciate the reviewer if he/she could share with us a reference paper for our future experiments.

We are really sorry that we could not clearly understand the reviewer comment “Silica gel column chromatography should not abbreviated unless they are clearly declared”. It is requested to please clearly elaborate what the reviewer mean that we could respond in a better way, thank you.

It was a mistyping error, the NMR spectrum in figure 3 are 1H and 13C NMR respectively. We correct all the errors related to the NMR, updated the figure and changed the figure legend and figure number respectively.

  1. Arora, S.; Itankar, P. Extraction, isolation and identification of flavonoid from Chenopodium album aerial parts. Journal of traditional and complementary medicine 2018, 8, 476-482.

Round 2

Reviewer 2 Report

The revised manuscript was checked and the reviewer realized that the authors made some efforts for improvement. However, there still remain some points to which the reviewer would like the authors to pay attention as the followings.
(1) The reviewer understands the authors' insistence that the main focus of the study is mainly related to the functional evaluation of OsF3H gene. Nevertheless, the reactions that the enzymes of interest work on are chemical conversion of specific functional groups. Therefore, the reviewer believes that chemical structures of three important molecules, which the authors insist are too famous to be drawn in the manuscript, should be clearly shown.
(2) There are still immature errors of English as listed below:
(2-1) In line 21, "increased" should be "increase".
(2-2) In line 22, "whish" should be "which".
(2-3) In line 84, "flavonone" should be "Flavonone".
(2-4) In line 181, "silica cylinder" should be "silica gel cylinder".
(2-5) In line 187, "a drier" should be "a blow dryer".
(2-6) In line 194, "fraction distillation" should be "fractionation".
(2-7) In line 196, "perform" should be "performed".
(2-8) In line 199, a period (.) is missing after "repeats".
(2-9) In line 202, 1 and 13 of 1H and 13C should be superscript.
(2-10) In line 203, "in MeOD solutions" should be "as methanol-d4 solutions".
(2-11) In line 203, 1 and 13 of 1H and 13C should be superscript.
(2-12) In line 255, "quercetin was appeared" should be "quercetin appeared".
(2-13) In line 256, the reviewer believes that "showing" would be better than "which shows".
(2-14) In line 257, "catalyzed naringenin into kaempferol" should be "catalyzed conversion of naringenin to kaempferol".
(2-15) In line 305, the reviewer believes that either "useful" or "powerful" would be better than "significant".
(2-16) In line 308, "1H and 13C NMR (Fig. 3A and B respectively)" should be "13C and 1H NMR (Fig. 3A and B, respectively)". Please do not make 1 and 13 superscript.
(2-17) In line 309, "analysis" should be "analyses".
(2-18) In line 312, the reviewer believes that "poorly" would be better than "slightly".
(2-19) In lines 319-320, the reviewer believes that "The identification ... kaempferol implied that ..." would be better than "With the identification ... kaempferol, it is predicted that ...".
(2-20) In lines 328-329, if the authors put the figures in this way, the reviewer would write the legend as follows: 13C and 1H NMR spectra (A and B, respectively) of quercetin. The proposed structure is shown in C.
(2-21) In line 333, "different pathways responsible for the synthesis of" should be "different pathways that are responsible for the syntheses of".
(2-22) In line 337, "quercetin which" should be "quercetin, which", judging from the context.
(2-23) In line 345, "the both the kaempferol and quercetin synthesis was not" should be "neither kaempferol nor quercetin synthesis was".
(2-24) In line 347, the reviewer believes that "increased kaempferol and quercetin significantly (p<0.05)" would be clearer than "significantly (p<0.05) increased kaempferol and quercetin".
(2-25) In line 362, the reviewer would remove "this is a novel finding and".
(2-26) In line 365-366, "kaempferol converts" should be "kaempferol is converted".
(2-27) In line 371, "which mean" should be "suggesting".
(2-28) In line 371, "also produces [40] which is most probably be due to" should be "also be produced [40], which is probably due to".
(2-29) In line 375, "naringenin converts" should be "naringenin is converted".
(2-30) In line 376, "enzymes specificity" should be "enzyme specificity".
(2-31) In line 378, "synthesis" should be "synthesizing".
(2-32) In line 381, "using" should be "used".
(2-33) In lines 382-383, "enzyme converts" should be "enzyme that converts".

Author Response

Response to Reviewer 2 Comments

The revised manuscript was checked and the reviewer realized that the authors made some efforts for improvement. However, there still remain some points to which the reviewer would like the authors to pay attention as the followings.
(1) The reviewer understands the authors' insistence that the main focus of the study is mainly related to the functional evaluation of OsF3H gene. Nevertheless, the reactions that the enzymes of interest work on are chemical conversion of specific functional groups. Therefore, the reviewer believes that chemical structures of three important molecules, which the authors insist are too famous to be drawn in the manuscript, should be clearly shown.

Response 1: According to the reviewer suggestion, we drawn the chemical structures of all the three important metabolites and presented as supplementary figure 3. Also, description incorporated into main text in line 367-368.

(2) There are still immature errors of English as listed below:

Response: Authors are very thankful to the reviewer for pointing out very critical mistakes. As the amendments are mostly single word change and we make the correction very carefully so in response we will only mention line number instead of simply writing the word “changed”.

(2-1) In line 21, "increased" should be "increase".

Response: line 21

(2-2) In line 22, "whish" should be "which".

Response: line 22

(2-3) In line 84, "flavonone" should be "Flavonone".

Response: line 84

(2-4) In line 181, "silica cylinder" should be "silica gel cylinder".

Response: line 181

(2-5) In line 187, "a drier" should be "a blow dryer".

Response: line 187

(2-6) In line 194, "fraction distillation" should be "fractionation".

Response: line 194

(2-7) In line 196, "perform" should be "performed".

Response: line 196

(2-8) In line 199, a period (.) is missing after "repeats".

Response: line 199

(2-9) In line 202, 1 and 13 of 1H and 13C should be superscript.

Response: line 202

(2-10) In line 203, "in MeOD solutions" should be "as methanol-d4 solutions".

Response: line 203

(2-11) In line 203, 1 and 13 of 1H and 13C should be superscript.

Response: line 203-204

(2-12) In line 255, "quercetin was appeared" should be "quercetin appeared".

Response: line 255

(2-13) In line 256, the reviewer believes that "showing" would be better than "which shows".

Response: line 256

(2-14) In line 257, "catalyzed naringenin into kaempferol" should be "catalyzed conversion of naringenin to kaempferol".

Response: line 257

(2-15) In line 305, the reviewer believes that either "useful" or "powerful" would be better than "significant".

Response: line 305

(2-16) In line 308, "1H and 13C NMR (Fig. 3A and B respectively)" should be "13C and 1H NMR (Fig. 3A and B, respectively)". Please do not make 1 and 13 superscript.

Response: line 308, 13C and 1H in figure legends were kept same as suggested by reviewer line 330-331.

(2-17) In line 309, "analysis" should be "analyses".

Response: line 309

(2-18) In line 312, the reviewer believes that "poorly" would be better than "slightly".

Response: line 312

(2-19) In lines 319-320, the reviewer believes that "The identification ... kaempferol implied that ..." would be better than "With the identification ... kaempferol, it is predicted that ...".

Response: line 319-320

(2-20) In lines 328-329, if the authors put the figures in this way, the reviewer would write the legend as follows: 13C and 1H NMR spectra (A and B, respectively) of quercetin. The proposed structure is shown in C.

Response:  we re-write the legend of figure 3 according to the reviewer suggestion line 330-331.

(2-21) In line 333, "different pathways responsible for the synthesis of" should be "different pathways that are responsible for the syntheses of".

Response: now line 335

(2-22) In line 337, "quercetin which" should be "quercetin, which", judging from the context.

Response: now line 339

(2-23) In line 345, "the both the kaempferol and quercetin synthesis was not" should be "neither kaempferol nor quercetin synthesis was".

Response: now line 347

(2-24) In line 347, the reviewer believes that "increased kaempferol and quercetin significantly (p<0.05)" would be clearer than "significantly (p<0.05) increased kaempferol and quercetin".

Response: now line 350

(2-25) In line 362, the reviewer would remove "this is a novel finding and".

Response: we removed “this is a novel finding and” now line 365.

(2-26) In line 365-366, "kaempferol converts" should be "kaempferol is converted".

Response: now line 370

(2-27) In line 371, "which mean" should be "suggesting".

Response: now line 375

(2-28) In line 371, "also produces [40] which is most probably be due to" should be "also be produced [40], which is probably due to".

Response: now line 376

(2-29) In line 375, "naringenin converts" should be "naringenin is converted".

Response: now line 379

(2-30) In line 376, "enzymes specificity" should be "enzyme specificity".

Response: now line 380

(2-31) In line 378, "synthesis" should be "synthesizing".

Response: now line 382

(2-32) In line 381, "using" should be "used".

Response: now line 386

(2-33) In lines 382-383, "enzyme converts" should be "enzyme that converts".

Response: now line 387.
